# Retinal Blood-Vessel Extraction Using Weighted Kernel Fuzzy C-Means Clustering and Dilation-Based Functions

**DOI:** 10.3390/diagnostics13030342

**Published:** 2023-01-17

**Authors:** Kittipol Wisaeng

**Affiliations:** Technology and Business Information System Unit, Mahasarakham Business School, Mahasarakham University, Mahasarakham 44150, Thailand; kittipol.w@acc.msu.ac.th; Tel.: +66-086-6393870

**Keywords:** retinal blood vessel extraction, Weighted Kernel Fuzzy C-Means Clustering, Dilation-Based Function, diabetic retinopathy

## Abstract

Automated blood-vessel extraction is essential in diagnosing Diabetic Retinopathy (DR) and other eye-related diseases. However, the traditional methods for extracting blood vessels tend to provide low accuracy when dealing with difficult situations, such as extracting both micro and large blood vessels simultaneously with low-intensity images and blood vessels with DR. This paper proposes a complete preprocessing method to enhance original retinal images before transferring the enhanced images to a novel blood-vessel extraction method by a combined three extraction stages. The first stage focuses on the fast extraction of retinal blood vessels using Weighted Kernel Fuzzy C-Means (WKFCM) Clustering to draw the vessel feature from the retinal background. The second stage focuses on the accuracy of full-size images to achieve regional vessel feature recognition of large and micro blood vessels and to minimize false extraction. This stage implements the mathematical dilation operator from a trained model called Dilation-Based Function (DBF). Finally, an optimal parameter threshold is empirically determined in the third stage to remove non-vessel features in the binary image and improve the overall vessel extraction results. According to evaluations of the method via the datasets DRIVE, STARE, and DiaretDB0, the proposed WKFCM-DBF method achieved sensitivities, specificities, and accuracy performances of 98.12%, 98.20%, and 98.16%, 98.42%, 98.80%, and 98.51%, and 98.89%, 98.10%, and 98.09%, respectively.

## 1. Introduction

Initially, human experts primarily used digital retinal images in ophthalmic clinics to identify DR and related eye diseases. The retinal blood vessels are important diagnostic indicators for DR and the pathologies of systems within the human eye. More and more research indicate that the accurate extraction of retinal blood vessels is helpful in the analysis of other related ophthalmic diseases. The manual extraction of blood vessels requires an expert ophthalmologist’s skills. Although such manual extraction is possible, it is time-consuming and there can be human error when working with large image datasets. Therefore, automated systems for the precise extraction of retinal blood vessels are urgently needed to reduce the workload of expert ophthalmologists. An example of the automatic extraction of blood vessels is illustrated in Figure 1.

Considering the typical retinal blood vessels and the background region information in the digital retinal images shown in Figure 1, three challenges make the retinal vessel-extraction task difficult:
Retinal images often require higher contrast and quality, making it difficult to extraction the Region of Interest (ROI).Most retinal images suffer from imbalanced illumination and noise (salt and pepper), making it difficult to distinguish the blood vessel from the background image.Retinal blood vessels come in many shapes, sizes, and unexpected forms, making it challenging to identify large and small blood vessels.


In these circumstances, research on vessel extraction from retinal images has attracted a large number of researchers.

Related works and state-of-the-art studies enlighten this paper’s proposed method, which is based on machine learning and deep-learning methods. A selection of studies on retinal vessel extraction from 2015 to 2022 was reviewed to ensure that the summary of extraction techniques is as up-to-date. In choosing the articles, keywords such as blood vessel classification, blood vessel extraction, retinal imaging, DR segmentation, exudate segmentation, and red lesion extraction were selected. I searched for articles from websites such as Web of Science, PubMed, Science Direct, and IEEE Xplore. This review includes 216 papers from Q1 to Q4 journals and international conferences. 

Image extraction generally involves classifying ROI using specific image intensities or geometric features to suppress the unwanted background. This way, only specific areas are extracted as retinal blood-vessel regions. Typically, the anatomy of the retina consists of observable characteristics, such as the optic disc, retinal blood vessels, and eye pathologies that appear in retinal images. In addition, the researchers used retinal images from various data sources in their analysis, captured at different times, resolutions, and intensities. Accordingly, determining whether retinal blood-vessel extraction achieves the same accuracy as that of an expert diagnosis is a challenging task that has attracted many researchers.

Recent implementations of deep-learning algorithms for retinal-image extraction were proposed by Soomro et al. [1]. Imran et al. [2] proposed the effectiveness of different deep-learning algorithms for automating blood-vessel extraction. Their proposed method falsely extracted the blood vessel on an abnormal image, and micro blood vessels were extracted due to the noise. Chen et al. [3] proposed deep-learning algorithms for retinal-image analysis. Their proposed method demonstrated significant efficacy in extracting retinal blood vessels. However, it had limitations in extracting thin blood vessels, especially in noisy and low-intensity images. A thorough analysis of conventional supervised, unsupervised, and neural network methodologies and statistics was presented by Khan et al. [4]. Jia et al. [5] analyzed deep-learning and traditional machine learning methods for retinal-image extraction. However, the ability of their approach to accurately extract a micro blood vessel was limited. Li et al. [6] proposed effective deep-learning techniques for extracting blood vessels. Next, the main focus of Badar et al. [7] was on deep learning techniques for retinal-image analysis. The main features of their proposed method were its improved extraction performance and reduced running time. The extraction of retinal blood vessels was automated using several supervised and unsupervised methods. The unsupervised method is the most prevalent technique for autonomously extracting the retinal blood vessel [8,9]. These methods can be extracted into three categories: matching filters [10,11,12], extraction based on blood-vessel tracing [13,14,15], and model-based extraction [16]. Since they cannot exploit the hand-labeled ground-truth image, unsupervised algorithms show some performance limitations. In contrast, unsupervised algorithms are trained using annotations and can take advantage of real datasets. In two steps, retinal blood-vessel extraction is carried out by supervised models: feature extraction and pixel classification. The methods can further break down features into features that are created manually or automatically. In machine learning, feature extraction from retinal images is done manually, and specific methods, such as the K-Nearest Neighbor (KNN) method [17] and the Support Vector Machine (SVM) method [18], are used. The generalization ability is lacking when features are manually selected because manual selection is application-specific and new features cannot be extracted [19]. 

For image processing, deep learning techniques, particularly Convolutional Neural Networks (CNNs), have received much attention [20,21]. Deep learning techniques use enormous amounts of data to learn features, while automatically minimizing human inference. They are superior at extraction because they can automatically learn multiple-level patterns and are not constrained by a particular system. The following issues are typically present in the proposed deep-learning techniques: (1) the model’s down-sampling factor is too high, causing numerous micro blood vessels to lose their feature information, which can never be recovered; (2) the DR and retinal blood vessels produce results that are low in accuracy; and (3) it is only possible to obtain accurate blood-vessel information from the extracted blood-vessel image with a great deal of noise. 

Jiang et al. [22] suggested a network based on the fully convolutional version of AlexNet. They used Gaussian smoothing to lessen the discontinuity between the optic disc and the replacement region. This method has the advantage of correctly affecting the retinal blood vessel even in cases of DR. In contrast, the application of this method is limited to situations where the retinal blood vessels are connected. Li et al. [23] built FCM with skip connections to improve retinal blood-vessel extraction. The extraction accuracy was increased by active learning, using fewer manually marked samples. The iterative training method improved the proposed model’s performance. This strategy has produced positive results with different datasets, but small datasets will yield even better results. 

In addition to proposing a fully convolutional network, Atli and Gedik [24] were the first to use up-sampling and downsampling to capture small and large blood vessels, respectively. Their suggested method used the STARE (Structured Analysis of the Retina) dataset for retinal blood-vessel extraction. Zhang and Chung [25] considered this multi-class extraction task and included an edge-aware technique. Five categories of pixels were created for background and micro blood vessels. As a result, the network was able to concentrate on the blood vessel boundary zones. To make optimization easier, they made use of deep supervision. However, their proposed method had limitations in accurately extracting blood vessels with a width of one to several pixels. Although their proposed method achieved accurate results, the overall sensitivity and specificity were low compared to conventional methods and can be improved by parameter tuning. 

Mishra et al. [26] suggested a straightforward U-net and introduced data-aware deep supervision to improve micro blood-vessel extraction. They determined the average input diameter of the retinal blood vessels and the layer-wise effect, adding further layers as needed. Receptive fields were used to identify layers that notably extracted aspects of retinal blood vessels. 

Deformable convolution was suggested by Jin et al. [27] for classifying blood vessels. The flexible convolution block learned offsets to modify the receptive fields, enabling it to capture blood vessels in various sizes and shapes. In comparison to U-net and the deformed convolution network [28], the proposed deformable U-net outperformed them on the DRIVE (Digital Retinal Images for Vessel Extraction), STARE, and CHASE (Child Heart and Health Study in England) datasets, as well as on two additional datasets: WIDE [29] and SYNTHE (Synthesizing Retinal and Neuronal Images) [30]. 

In order to extract micro blood vessels, Dharmawan et al. [31] presented a combined Contrast-Limited Adaptive Histogram Equalization (CLAHE) method with a novel match filter. They built the new matching filter on the multiscale and modified the Dolph–Chebyshev type I function. Their proposed combination improved extraction performance by using a preprocessing step, and the preprocessed image was fed directly to the fine extraction process. Compared to standard CLAHE, their method discovered more micro blood vessels, although it still committed many extraction errors. Dilated convolution was also used to extract blood vessels to increase the receptive fields [32]. Lopes et al. [33] also examined the effects of a series of downsampling methods, such as max-pooling, convolution with a 2-by-2 kernel, and convolution with a 3-by-3 kernel. Their method can help address low-intensity images and micro blood-vessel extraction in cases of DR disease. According to Soomro et al. [34], they achieved superior outcomes when employing convolution as a downsampling operation. They used morphological reconstruction during the post-processing stage to eliminate the small pixels in the extracted images. A black ring surrounds the Field of Vision (FOV) in retinal imaging. Networks should pay closer attention to the FOV because there is no information in the black ring. This method can more effectively identify normal and abnormal images but requires a mathematical morphology method to extract the micro blood vessels in the extracted image. 

The region of interest has been identified and feature representations have been strengthened in blood-vessel extraction via the attention mechanism [35]. Luo et al. [36], Lian et al. [37], and Lv et al. [38] manually built attention masks that were the same size as the retinal images to detect the region of interest. Li et al. [39], Li et al. [40], and Fu et al. [41] provided only a few examples. According to Tang et al. [42], their created attention modules and networks learned their attention mappings, rather than relying on experts. Although this method cannot segment micro blood vessels, the process may further improve the accuracy of results in the future. According to other works, blood-vessel boundaries or micro blood vessels were extracted independently before being combined to create a complete extraction [43,43,44]. Other researchers proposed segmenting data from coarse to fine by cascading different subnetworks [45]. The original images and the extracted output from previous sub-models were used as the input for the next sub-model. 

When using deep learning to extract retinal blood vessels, the following issues have been reported:
There is a need for training samples with clear labels. Even though there are many retinal images, collecting annotated data is quite challenging because it involves qualified experts, takes a lot of time, and is expensive.The current retinal image samples are of low quality. As a result, deep learning models cannot develop more robust feature representations. The performance of the proposed methods is hampered by image noise, low intensity, and various features of DR diseases.There is an issue concerning training sample class imbalance. The performance of networks is harmed by disparity in the amount of positive and negative training examples. Class imbalance affects large and micro blood vessels, DR, and backgrounds. Because there are more non-blood-vessel pixels than blood-vessel pixels, deep learning models frequently categorize pixels in boundaries as non-blood-vessel pixels.Since the error extraction of micro blood-vessel pixels has less impact on the overall loss, the network performs poorly on micro blood vessels than on large blood vessels.


This research focuses on retinal blood-vessel extraction, with an emphasis on an extraction method that does not require specialized hardware for training the developed methods, decreases computational time, eliminates hyper-parameter adjustment, and minimizes memory. I chose the WKFCM-DBF method because of its many advantages. WKFCM-DBF identifies retinal blood vessels without a ground-truth image. This is a great advantage, as large datasets are typically unavailable for large-scale screening programs. In addition, Dilation-Based Functions and optimal thresholding are based on attributes of micro blood vessels in retinal images.

The remaining portions of the article are divided into the following sections. A detailed description of the data preparation, image normalization, color space selection, noise removal, and Optic Disc (OD) feature extraction is provided in Section 2. Section 3 provides a discussion of the proposed WKFCM-DBF method together with discussion of the standard FCM and WKFCM methods. It also covers method-training procedures and test setup. The experimental results, materials, evaluation criterion, robustness of extraction, and comparison of the proposed WKFCM-DBF method with state-of-the-art methods are provided in Section 4. Finally, the conclusion is drawn in Section 5.

## 2. Data Preparation

This section details a proposed method for extracting retinal vessels. Figure 2 shows a diagram of the proposed WKFCM-DBF method.

I separated green, red, and blue channels from the RGB retinal image and then selected the green image. I identified the following crucial factors as being essential for accurate and reliable retinal blood-vessel extraction:
The original image contrast was enhanced using preprocessing techniques, such as separating the RGB image to the green channel component, normalizing the colors with averaging threshold optimization, and applying the median filter on the improved image to decrease noise.A morphological Dilation-Based Function was applied to maximize retinal blood-vessel detail. Then, I used optimal global thresholding to improve binary image quality.Using images enhanced for accurate extraction allows well-developed algorithms to extract more retinal blood-vessel details. Because they have more shapes, positions, and sizes, I was able to extract them more effectively. Additionally, micro blood-vessel connections support the reuse of low-quality images in unseen images. This is important for extraction. The accuracy of specific proposed methods was also very high. Augmented images based on morphological Dilation-Based Function are a viable alternative to improve the performance of micro blood-vessel extraction and collect more spatial information while preserving the running time of the extraction process. I selected suitable clusters of WKFCM parameters to acquire images of large and micro blood vessels.Using a suitable technique, I was able to pay more attention to retinal blood vessels, especially micro blood vessels, if the correct parameters were used. I used enhanced images and a dilation function to address imbalance problems, resulting in binary images via different parameters. The optimal parameter thresholds (shape, OD, and location) were empirically examined and applied to remove non-blood-vessel components. With this approach, the shape of retinal blood vessels can be better visualized to help expert ophthalmologists diagnose disease.The proposed WKFCM-DBF method was evaluated using three different datasets after training samples from each dataset. I was able to carry out cross-validation for additional validation. I tested the proposed method on the unseen or private dataset. The preprocessing steps, including color normalization with averaging threshold optimization and application of the median filter to remove noise in enhanced images to enhance image quality, are detailed below.


### 2.1. Image Normalization Using Averaging Threshold Optimization

Image enhancement is an essential preprocessing step for improving details in the retinal images. A simple averaging threshold optimization technique enhances image contrast and maintains the average brightness [46]. The histogram distribution of the original image is uneven in terms of image brightness. The brightness of the center of the image and the brightness of the periphery of the image are significantly different (see Figure 3b). Therefore, a non-uniform brightness distribution of the dark areas, as shown in Figure 3b, is modified by adjusting the weight of the histogram in such areas to the standard brightness by increasing the image intensity, while maintaining the mean brightness of the original retinal image (see Figure 3d). The averaging threshold optimization technique is described below.

The image threshold is calculated via Equation (1).
(1)$$\bar{h}(i)\leftarrow \frac{h(i)+\tau \times max\{h(i)\}}{2},0<\tau <1$$
where max{h(i)} is the highest level of intensity within the image i = 0, …, L − 1. To simultaneously achieve the two objectives of improving the original image in the dark area and maintaining the original image brightness, the averaging threshold τ, derived via Equation (1), is proposed to enhance the image brightness using the average threshold optimization method. An objective function for optimization is defined from the information content to determine the optimal averaging threshold within the image to increase the brightness of the original image. Therefore, to meet the requirements of average optimization, I applied the optimal threshold value to improve the brightness in the dark areas, using Equation (2):(2)$$ℑ=\frac{\left|{\hat{I}}_{average,enhanced}-I_{average}\right|}{I_{average}}\times H$$
where Î_average, enhanced_ is the average brightness of the enhanced image Î_enhanced_. The variable H represents the entropy of the maximized value of the information provided by Equation (3):(3)$$H=-\sum_{i=0}^{L-1}p(i)logp(i),p(i)\hat{h}(i)/N$$

The objective function ℑ is then maximized. A final improved image Î_enhanced_ is obtained using the averaging threshold optimization technique, with brightness improved by 0.58 as illustrated in Figure 4.

### 2.2. Color Space Selection and Noise Removal

The second preprocessing stage is the selection of suitable color channels for the extraction of retinal blood vessels. Figure 5 shows the color space extracted from the input image, i.e., the red, green, and blue channels. In this paper, I chose the green image because this color channel shows the detail of blood vessels and retinal information more than other channels do (see Figure 5c). Then, I used the default parameters of a median filter to suppress the noise in the improved image. The presence of noise in the retinal images was also removed to obtain a smooth image that was suitable for extracting the presence of the blood vessels.

### 2.3. OD Feature Extraction Using Top-Hat Transform

As shown in Figure 5c, the intensities of OD and DR are typically greater than that of the retinal vessels. Therefore, the bright pixels are subtracted from the dark regions. In this stage, I used the mathematical morphology based on top-hat transform to redistribute the grayscale intensity from a normalized retinal image to generate a feature of a blood-vessel and non-blood-vessel extraction. Basic mathematical morphology top-hat operators are defined based on two types: White Top-Hat transform (WTH) and Black Top-Hat transform (BTH) [47]. WTH extracts blood-vessel regions (bright image features) and BTH extracts background regions (dark image features). Hence, I applied the WTH operator to extract optic disc regions and small features from retinal images. The extracted blood-vessel image features based on the mathematical morphology of a WTH can be defined as in Equation (4):(4)WTH(f)=f−f∘b
where WTH represents the white top-hat transform, where the bold vessel features are highlighted more brightly than the other surrounding features, f denotes the original grayscale image, b is the grayscale structuring element, and ◦ represents the opening operation. Accordingly, the WTH only strengthens the retinal blood vessels, suppressing intensity variation due to possible DR and OD. The top-hat modified image is shown in Figure 6, where the background and likely DR are smoothed and blood vessels are more prominent than the OD. According to Figure 6, the grayscale intensity is selected as the blood-vessel region for extraction (Figure 6a), in which approximately 0.65 pixels (see Figure 6b) have an intensity of 0.1 after redistribution (see Figure 6d). A demonstration of the intensity changes before and after the transform is shown in Figure 6c). Meanwhile, the histogram between intensity values 0.1 and 0.6 implies that a possible threshold exists to distinguish the blood-vessel and non-blood-vessel regions.

## 3. Proposed Method

The object of retinal blood-vessel extraction is to segment large and micro vessels as much as possible. Based on the work of image preprocessing, the first step in this section was to apply the standard FCM method to extract retinal blood vessels. After that, I found that the optimal function for each extraction stage became the subsequent critical optimization. The optimization function of blood-vessel extraction based on the WKFCM-DBF method was investigated in that part. To train and test images for the extraction stage, preprocessed images, as described in Section 2.1, were used to train and test the images.

### 3.1. The Standard FCM Algorithm

FCM achieves clustering by repeatedly searching for a set of fuzzy clusters and their corresponding cluster centers that represent the structure of the data. The clustering algorithm depends on the user to indicate the number of clusters present in a dataset to be clustered. Let c be the number of fuzzy clusters (1 < c < n) and X = {x_1_, x_2_, …, x_n_} be the number of examples provided. Then, c fuzzy clusters are provided with a probabilistic cluster partition of X by minimizing the within-cluster sums of the squared error objective function. The image extraction algorithms based on the FCM analysis are fuzzy optimization algorithms [48,49]. This study investigated improvements to traditional FCM algorithm in image extraction. The following objective functions were utilized, as determined via Equation (5), to enhance the method [50].
(5)$$J_{m}(U,V)=\sum_{i=1}^{C}\sum_{k=1}^{N}{\left(u_{ik}\right)}^{2}{\left(d_{ik}\right)}^{2}$$
where J_m_(U, V) represents the sum of squared error criteria, its minimization produces fuzzy clusters by the membership matrix U, V is the set of cluster centers, C represents the total number of clusters, N denotes the total number of objects in any gray image in the dataset, d_ik_ represents the distance measure between object k and cluster center i, and u_ik_ represents the degree of membership of kth pixel belong to the i^th^ cluster that satisfies the condition $\sum_{i=1}^{C}u_{ik}=1\;and\;\forall u_{ik}\in \left[0,\ldots ,c\right]$. The commonly used standard FCM is the basis of the clustering algorithms [51,52]. The objective function of standard FCM is calculated as in Equation (6).
(6)$$J_{m}=\sum_{i=1}^{C}\sum_{k=1}^{N}u_{ik}^{m}{\left\|x_{k}-v_{i}\right\|}^{2}$$
where x_k_ (k = 1,2,…,N) represents the k^th^ pixels vector of pixels of a gray level image, v_i_ {v_i_ = 1, 2, …, C} represents the i^th^ cluster for i = 1, 2, …, N, u_ik_ is the fuzzy membership degree of pixel x_i_ in cluster k, m represents the index of the fuzzy weight (the fuzzifier parameter of the algorithm), ||…|| represent some inner product induced norm, and ||x_k_ − v_i_||^2^ is the squared distance between the data element x_k_ and the cluster center v_i_ (generally defined by the Euclidean distance) [53]. The values of membership (u_ik_) and the new cluster centers (v_i_) are defined as Equation (7).
(7)$$u_{ik}=\frac{{\left({\left\|x_{k}-v_{i}\right\|}^{2}\right)}^{-1/\left(m-1\right)}}{{\sum_{j=1}^{C}\left({\left\|x_{k}-v_{i}\right\|}^{2}\right)}^{-1/\left(m-1\right)}},v_{i}=\frac{\sum_{k=1}^{N}u_{ik}^{m}x_{k}}{\sum_{k=1}^{N}u_{ik}^{m}K}$$

However, the traditional FCM is essentially a local random search clustering algorithm based on Gradient Descent (GD) and thus has a greater dependence on the initial conditions.

### 3.2. The WKFCM Method

The WKFCM method has been derived from the traditional FCM based on modifying the kernel function, the membership degrees, and the fuzzy cluster optimization described in the following discussion [49,54,55]. The kernel function of FCM is calculated in Equation (8).
(8)$$K\left(x,y\right)=\left(\Phi \left(x\right),\Phi \left(y\right)\right)$$
where (.) represents the inner product. In this study, the vector is modified using the Gaussian kernel function used in Equation (9) [56].
(9)$$K\left(x,y\right)=exp\left(-{\left\|x-y\right\|}^{2}/\sigma^{2}\right)$$
where σ represents the kernel function’s characteristic parameter. The objective function for the WKFCM extraction algorithm is provided in Equation (10).
(10)$$J_{m}=\sum_{i=1}^{C}\sum_{j=1}^{N}u_{ij}^{m}{\left\|\Phi \left(x_{k}\right)-\Phi \left(v_{i}\right)\right\|}^{2}$$
where Φ represents a non-linear function that maps a low-dimensional feature space to a high-dimensional feature space.${\left\|\Phi \left(x_{k}\right)-\Phi \left(v_{i}\right)\right\|}^{2}$ in Equation (10) is given by Equation (11):(11)$${\left\|\Phi \left(x_{k}\right)-\Phi \left(v_{i}\right)\right\|}^{2}=K\left(x_{k},x_{k}\right)+K\left(v_{i},v_{i}\right)-2K\left(x_{k},v_{i}\right)$$

The objective function of Equation (11) can be expressed by Equation (12).
(12)$$J_{m}=2\sum_{i=1}^{C}\sum_{k=1}^{N}u_{ik}^{m}\left(1-K\left(x_{k},v_{i}\right)\right)$$

Then, the cluster centroids v_i_ and the new fuzzy membership-degree value u_ik_ are calculated using Equations (13) and (14).
(13)$$v_{i}=\frac{\sum_{k=1}^{N}u_{ik}^{m}K\left(x_{k},v_{i}\right)x_{k}}{\sum_{k=1}^{N}u_{ik}^{m}K\left(x_{k},v_{i}\right)}$$
(14)$$u_{ik}=\frac{{\left(1-K\left(x_{k},v_{i}\right)\right)}^{-1/\left(m-1\right)}}{\sum_{j=1}^{C}{\left(1-K\left(x_{k},v_{j}\right)\right)}^{-1/\left(m-1\right)}}$$

The remaining steps and calculations of the WKFCM are the same as those of the standard FCM. I formalized the WKFCM algorithm as follows:
The data points were normalized so that a hypercube bounds them. Given the data set Z, I set the number of clusters (c) at 2, the weighting exponent (m) in the FCM at 3, and the tolerance of termination (T) at 120.The clusters of centroids were computed by the standard FCM method.The clusters of centroids (step 2) as were regarded as input to the WKFCM.The new membership values of the data points were computed by the WKFCM method, using Equation (14).The value of ∆ was calculated, using Equation (15):(15)$$\Delta =\left\|U^{{}_{\left(l+1\right)}}-U^{{}_{\left(l\right)}}\right\|$$
where l represents the iteration-counting parameter. If ∆ > T, repeat steps 5, 6, and 7. Otherwise, stop at some iteration.The clusters of centroids were updated by WKFCM, using Equation (13).The new membership values were updated by WKFCM, using Equation (14).


I applied the preprocessing method to all datasets, which had been normalized and enhanced for the coarse extraction stage. Then, I used the WKFCM for 9 (m = 1, 2, 3, 5, 7, 9, 11, 12, 13) with different random initialization processes of the membership values of an object to all classes in the dataset. For random initialization for the kernel function, I used three kernel function values (σ =0.3, 0.5, 0.7) of the adjustable parameter, σ, of the Gaussian kernel function to find the optimized result out of the three. Next, I provided the optimized values of the number of clusters for five different validity measures (c = 1, 2, 3, 4, 5) for the best choice among five values of the WKFCM obtained in 150, the maximum number of iterations. Finally, an initial threshold value (T) of 120 was selected as the minimum value to classify the histograms between the two classes. The blood vessel extracted using the proposed WKFCM method is illustrated in Figure 7. Three retinal images from the DRIVE, STARE, and DiaretDB0 databases are displayed in Figure 7.

### 3.3. Improved WKFCM

While extracting blood-vessel structure, a large and micro-vessel extraction will also run simultaneously with the same input image. However, the input image is converted to binary pixels to extract blood vessels from background pixels. As mentioned, the phase will be the retinal blood vessel in white pixels and the background with black pixels. During the coarse extraction, I found the loss of micro-vessel pixels. Therefore, I preserved the micro vessel while maintaining a computation time and superimposing their results in the original images. The connectivity between the current pixel and every one of its surrounding neighbors was examined using a morphological dilation operator. If connected, I categorized them into the same group. Accordingly, I adopted the morphological dilation operator in this stage, instead of the WKFCM false extraction, using Equation (16).
(16)$$A⊕B=\left\{\left.z\right|{\left(\hat{B}\right)}_{z}\cap A\neq \phi \right\}$$
where A indicates dilation by B, ϕ represents the empty set, B represents the structure element, and $\hat{B}$ is the reflection of image B. Dilation operations are described as structuring element B on image A and moving it across the image, similarly to convolution. This structural component chooses exactly the input image and will be affected by the dilatation. To improve the WKFCM method, I adopted the DBF and optimal global thresholding that compared the extracted image with the ground-truth image by pixel-based evaluation. The DBF makes the extracted image more accurate than does the WKFCM method. In this stage, the structuring element applied to a binary image can be dilated as small-vessel pixels, and each structure element has a value of 7. Note: I retained all the “black” pixels in the original image with a dilation function and filled small holes. The proposed method utilizes optimal intensity thresholding after the WKFCM procedure of the grayscale image, through which I will generate blood vessels and background pixels according to Equation (17).
(17)$$g(vessel,background)=\left\{\begin{array}{l} 255\;if\;f(vessel,background)\geq 128\\ 0\;if\;f(vessel,background)<128 \end{array}\right.$$
where f(vessel, background) corresponds to the input image, g(vessel, background) denotes the binary pixel after thresholding, and 128 is the optimal value to classify the histograms between the two classes. Any pixel (vessel, background) in the image at which f(vessel, background) ≥ 128 is extracted as a vessel pixel; otherwise, the pixel is regarded as a background point or a non-vessel pixel. The extraction result is displayed in Figure 8, where pixels with intensity values greater than 128 (retinal blood-vessel pixels) are shown in white against a background of black. As shown on the left side of Figure 8a, there are a few false extractions; they segment the actual retinal blood vessels (white pixel) as the background (black pixel). The Dilatation-Based Function is added to optimal thresholding functions to reduce the false negative value (see Figure 8b).

### 3.4. Post-Processing

The final stage of blood-vessel extraction is post-processing, in which images overlap between the extracted and the ground-truth image. Here, overlap means combining the extracted and the ground-truth images into one by simply applying the mapping coefficient to both. In this stage, the Mapping Coefficient (MAPC) is a statistical indicator that can determine the similarity between ground-truth and extracted images. MAPC is calculated via Equation (18).
(18)$$MAPC=\frac{\left|GT\cap SM\right|}{\left|GT+SM\right|}$$
where the extracted image’s pixel value is |SM| and the ground-truth image’s pixel value is |GT|, while |GT∩SM| represents the common elements between the ground-truth and the extracted images, and |GT + SM| represents the total number of pixels in the image. The addition of the proposed improvement of the WKFCM method to the extraction, based on the morphological dilation functions and optimal global thresholding, is displayed at the right of Figure 9. I confirmed that false extraction values were reduced and proposed a more accurate image of vessel extraction.

## 4. Experimental Results

The following subsections provide a performance analysis by comparing the proposed preprocessing techniques, a novel WKFCM-DBF method, and state-of-the-art algorithms. To validate the effectiveness of the proposed method for extracting blood vessels, I compared my approach with those of previous studies.

### 4.1. Materials

Many publicly available retinal datasets detail retinal anatomy and blood vessels. This is essential in retinal analysis and blood-vessel extraction for training and testing algorithms on retinal databases. I evaluated the extraction method using three publicly available datasets:
DRIVE: A collection of retinal images from the Netherlands, covering a wide age range of patients [17].STARE: A collection of 80 retinal images from the United States [57].DiaretDB0 (Standard Diabetic Retinopathy Database Calibration Level 0): 130 retinal images from Kuopio University Hospital [58].


I used the retinal image of the DRIVE dataset from a DR screening program in the Netherlands. There were 400 diabetic patients in the screening program, ranging in years of age from 25 to 90. Forty retinal images were randomly selected from 400 diabetic patients, of which 7 images showed symptoms of DR and the remaining 33 images did not show any signs of DR. Each image was captured in JPEG format using a Canon CR5 non-mydriatic camera with a field of view of 45 degrees, 8 bits per color channel, and 768 × 584 pixels. There was a total of 40 images in DRIVE’s training and test sets. This database has two manual classifications; I used 12 images as training ground truth and the remaining 28 images to compare the proposed extraction method with manual extraction by an experienced ophthalmologist. 

STARE was conceived in 1975 by Michael Goldbaum, M.D., of the University of California, San Diego. The STARE dataset has 80 images corresponding to the ground-truth images used for blood-vessel extraction, 49 of which are normal images, while the remaining 31 images show various types of abnormal disease. The first expert manually marked retinal images in large blood vessels, while the second and third experts marked micro blood vessels. Extraction results are commonly used as a ground-truth image for computing performance. Each image was captured in PPM format using a TOPCON TRV-50 camera with a field of view of 35 degrees, 605 × 700 pixels, and 8 bits per color channel. 

Finally, I obtained the retinal image of the DiaretDB0 from Kuopio University Hospital. The screening population consisted of 130 color retinal images, of which 110 showed symptoms of DR and the remaining 20 did not show any DR. Each image was captured in JPEG format with a field of view of 50 degrees, 1500 × 1152 pixels, and 8 bits per color channel. The information from the three selected datasets is summarized in Table 1.

### 4.2. Performance Measurement

To describe the performance of the blood-vessel extraction method, more than an accuracy performance is needed to determine the quality of extraction methods. In this section, I applied mathematical metrics that are commonly used to measure the performance of blood-vessel extraction algorithms: sensitivity, specificity, and accuracy. The extracted binary vessel image was compared pixel-to-pixel with its corresponding ground-truth image from the test set to calculate the extraction algorithm with sensitivity, specificity, and accuracy. The retinal vessel that was extracted was converted to a binary image to distinguish the vessels from the retinal background. A ground-truth image, manually marked by an experienced ophthalmologist, was used to evaluate the performance of the vessel-extraction method. Four parameters—TP (True Positive), TN (True Negative), FP (False Positive), and FN (False Negative)—indicated that proposed method correctly and incorrectly extracted the blood vessel image with the ground-truth image marked by expert ophthalmologists.
TP represents vessels correctly extracted by the proposed method.FN represents retinal vessels extracted as non-vessels by the proposed method.TN represents non-vessels correctly extracted by the proposed method.FP represents non-vessels incorrectly extracted as vessels by the proposed method.


In this theory, sensitivity, specificity, and accuracy can be calculated via Equations (19)–(21).
(19)$$Sensitivity=\frac{TP}{TP+FN}\times 100$$
(20)$$Specificity=\frac{TN}{TN+FP}\times 100$$
(21)$$Accuracy=\frac{TN+TN}{\left(TP+TN\right)+\left(FP+FN\right)}\times 100$$

Sensitivity (SEN) represents the correctly extracted blood vessels as blood-vessel pixels by the proposed method. Specificity (SPEC) represents the accurately extracted non-blood-vessel as non-vessel pixels by the proposed method. At the same time, accuracy (ACC) represents the proportion of correctly extracted pixels as blood-vessel or non-blood-vessel pixels.

### 4.3. Analysis of Proposed Blood-Vessel Extraction

The proposed method was simulated within a MATLAB environment (The Mathworks, Inc.) and executed on a personal computer with an Intel (R) Core (TM) i7-6700K CPU at 4.00 GHz and 8 GB DDR3 RAM. To confirm the proposed method in retinal vessel extraction, I visualized and compared the activations from the different stages presented. Figure 9 shows a visual comparison of three extracted images from the DRIVE, STARE, and DiaretDB0 datasets by the proposed method. The parameters provided in Equations (19)–(21) were used for the proposed method. Their average was determined by applying the traditional FCM and the proposed WKFACM-DBF method with a non-enhanced and an enhanced image to 167 test images of the DRIVE, STARE, and DiaretDB0 datasets, as shown in Table 2 and Table 3, respectively.

As previously mentioned, the proposed method has been developed based on the data from a publicly available dataset. Therefore, even though this is a machine learning implementation, the extraction method is still biased for the different datasets. The proposed method with preprocessing performs better on the STARE dataset than on the DRIVE and DiaretDB0 datasets. The results for the DiaretDB0 dataset indicate that its accuracy was the lowest among all datasets, implying that its ability to correctly classify retinal blood vessels is not as good as that of the other datasets. Nevertheless, the accuracy of the DiaretDB0 dataset is still high (98.09%). Below is an illustration of a detailed analysis of the numerical results presented in Table 2.
If the proposed WKFACM-DBF method is used for non-preprocessed images or images with noise or artifacts, the value of SEN and SPEC may be decreased.If the proposed WKFACM-DBF method is applied to preprocessed images or noise-free images, the application of the proposed method results in higher accuracy than that of the non-preprocessing alternative. However, these results show that the proposed method performs best in the studied retinal blood-vessel extraction method, even when the noise level is high. However, as it is more effective on unseen images and the different datasets free of noise, the proposed method performs as well as the training dataset.


Next, the proposed WKFACM-DBF method was compared to the traditional FCM method and tested on the DRIVE, STARE, and DiaretDB0 databases. From Table 3, the accuracy of the proposed method on the STARE dataset has increased to 26.99% instead of 71.52% by the traditional FCM method, which implies the increased accuracy in retinal blood vessel image extraction.

The following is an illustration of a detailed analysis of the numerical results presented in Table 3:
The proposed WKFACM-DBF method consistently outperforms standard FCM algorithms on these three measures. Note: the significant improvement in ACC scores over scores with standard FCM indicates that the proposed algorithm can better segment more micro blood vessels. This demonstrates, in particular, that the proposed improvement method highlights blood vessels of various widths.The kernel functions technique has a good extraction ability and has been improved into the fuzzy function in the WKFACM-DBF method. The optimal number of clusters and fuzzy weighting can improve the clustering accuracy. Therefore, the WKFACM-DBF method performs better than the traditional FCM algorithm in extracting retinal blood vessels with similar structures.


### 4.4. Extraction Analysis with Private Dataset

The performance of the WKFACM-DBF method on the private dataset was analyzed. The private dataset contains 1800 retinal images (750 × 750 pixels) with a 45-degree field of view. Of all the retinal images, 200 contained DR lesions. The training set (30%) and the test set (70%) were divided. Each image in the training set had only one manual annotation, while the test set had three manual annotations provided by three experts. I used the same evaluation methodology as that of the publicly available datasets and used the ground-truth image annotated by three experts for performance evaluation. I compared the performance of the WKFACM-DBF method using the same parameters. Table 4 displays the pixel-based evaluation of the WKFACM-DBF method and optimal global thresholding with the non-enhanced and enhanced images for large and micro blood-vessel segmentation.

In the private dataset experiment, the proposed method obtained 95.38%, 95.60%, and 95.42% for SEN, SPEC, and ACC, respectively. The proposed method achieved slightly lower SEN, SPEC, and ACC. However, incorporating the preprocessing approach and the WKFACM-DBF method with an optimal parameter improved the technique’s overall performance.

### 4.5. Performance Comparison and Analysis

The performance comparison with other algorithms was carried out via a single-dataset test (the DRIVE dataset). The proposed WKFACM-DBF method does not have a training process and does not necessarily have single or cross-dataset tests. However, to fairly compare the performance of the proposed WKFACM-DBF method with other methods, I compared the results from the proposed method with the results from the same dataset test of the other methods. Each method’s best SEN, SPEC, and ACC values are highlighted in bold in Table 5, denoting the best extraction results. As shown in Table 5, it is pertinent that, on the DRIVE dataset, the SEN, SPEC, and ACC values of the WKFACM-DBF method are better than those of the state-of-the-art algorithms. It can also be observed in the same table that the ACC value offered is the best value among the traditional clustering methods. The sensitivities of [22] and my method are the first- and second-best, respectively, while the algorithm of [24] is the third-best among all extraction methods. As the best, second-best, and third-best deep learning algorithms, respectively, refs. [23,24,40] all have higher specificity values than my method; mine is only slightly lower. However, the average accuracy of my proposed method lies in first place as a result of achieving high accuracy in sensitivity and specificity.

The proposed WKFCM-DBF method outperforms state-of-the-art methods based on pixel-based evaluation metrics, including SEN, SPEC, and ACC (most of the time) on three published datasets, DRIVE, STARE, and DiaretDB0, respectively. A careful preprocessing step of the proposed method’s activation helped prove the hypothesis and increase the proposed method’s efficiency by preserving retinal quality information in the extraction stage. This critical hypothesis was validated based on the SEN, SPEC, and ACC metrics. Moreover, the average time required to process one image on a computer with 4.00 GHz Intel(R) Core (TM) i7–6700K CPU, 8GB (RAM) under the Microsoft Windows 10 32–bit operating system for all datasets was 4 s per image. The similar execution times for all datasets were due to the normalized and resized sizes of the original images before a feed to the extraction method. As a result, the WKFCM-DBF method was computationally efficient and fast compared to many state-of-the-art techniques. Table 6 shows the comparison results for the DRIVE, STARE, and DiaretDB0 datasets. The extraction accuracy of the STARE databases was exceptionally high, which implied the extraction success of the proposed method. This remarkable achievement proves the validity of a of complete preprocessing exercise to enhance retinal images before transferring the enhanced images to a novel blood-vessel extraction method by a three-combination stage model.

## 5. Conclusions

This study focused on developing a novel method that applies to blood-vessel extraction contaminated with noise or artifacts. I used a new hybrid clustering method for retinal vessel extraction. This paper’s contribution is mainly related to three steps. I first described a normalization based on the averaging threshold optimization technique by conducting a rigorous analysis of the proposed method. Next, I described a noise-reduction algorithm based on a median filter method. Then, the images from the preprocessing stage were improved, contributing to a more reliable extraction of blood vessels and consequently improving the method’s overall performance. I developed the retinal blood-vessel extraction method based on the traditional FCM method and its improvements. The developed method focuses on the fast extraction of blood vessels using Weighted Kernel Fuzzy C-Means (WKFCM) Clustering to draw the vessel feature from the retinal background. I enhanced the membership degree and the kernel employed in the objective function of the traditional FCM clustering approach to address its disadvantages. Then, the method focused on the accuracy of full-size images for regional vessel feature recognition of large and micro blood vessels to minimize false extraction. This method implemented the mathematical dilation operator from a trained model called Dilation-Based Function (DBF). Finally, an optimal parameter threshold was empirically determined to remove non-vessel components in the binary image and improve the overall blood-vessel extraction results.

The new WKFCM-DBF method was thoroughly evaluated on both actual normal and abnormal images, and based on SEN, SPEC, and ACC scores, I obtained compelling results compared with existing methods. Compared to the state-of-the-art algorithms, the WKFCM-DBF method’s main advantages are better accuracy in retinal blood-vessel extraction of either poor or high quality and less running time. Moreover, based on the images and the table in Section 4.5, the proposed method has better extraction results and certain advantages in pixel-based evaluation metrics. However, the method also has drawbacks. For a retinal image with DR intensity inhomogeneity, the algorithm is unable to provide perfect blood-vessel extraction results. This could be because the extraction results of the WKFCM-DBF method for blood-vessel images with DR intensity inhomogeneity could be better, as the method does not consider the features of DR intensity of the retinal images. In the future, this situation should be improved and the OD and DR lesions should be removed before the coarse segmentation stage.

## Figures and Tables

**Figure 1 diagnostics-13-00342-f001:**
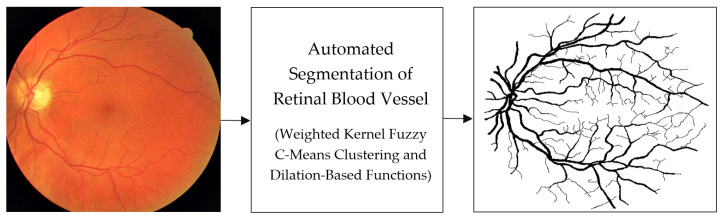
Example of the automated segmentation and extraction of blood vessels using Weighted Kernel Fuzzy C-Means Clustering and Dilation-Based Function.

**Figure 2 diagnostics-13-00342-f002:**
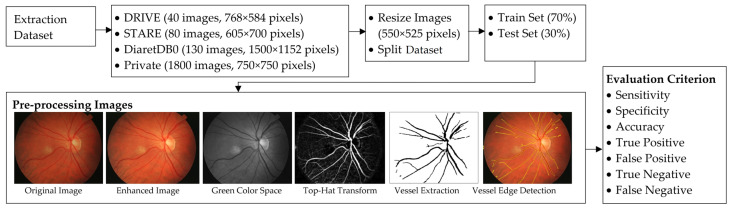
A diagram of the suggested extraction technique for the retinal blood vessels.

**Figure 3 diagnostics-13-00342-f003:**
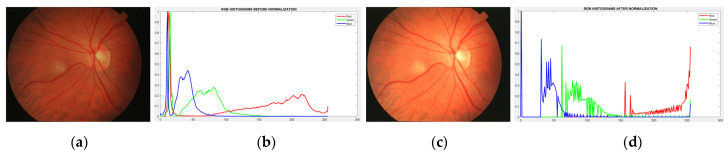
Demonstration of the color normalization process: (**a**) original retinal image, (**b**) corresponding histogram of (**a**), (**c**) normalized image, (**d**) histogram after normalized image.

**Figure 4 diagnostics-13-00342-f004:**
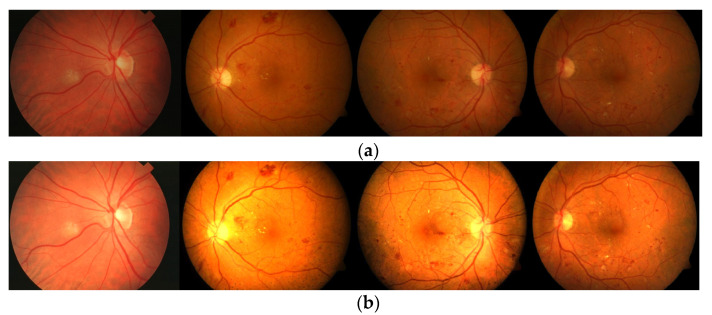
Test images: (**a**) original images and (**b**) normalized images by using the averaging threshold optimization technique.

**Figure 5 diagnostics-13-00342-f005:**
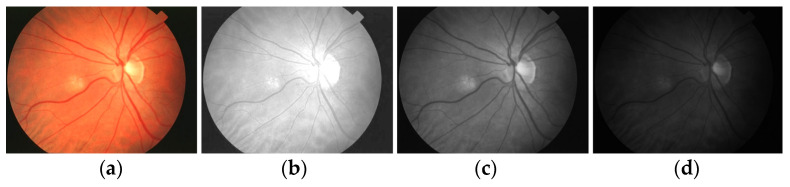
Retinal color space separation: (**a**) enhanced image, (**b**–**d**) RGB and individual space, respectively.

**Figure 6 diagnostics-13-00342-f006:**
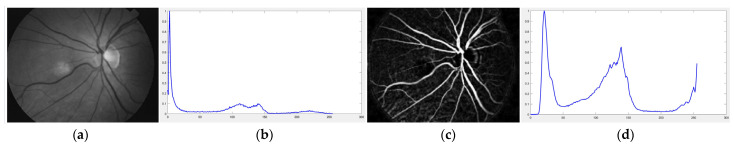
Example of the retinal image smoothed using mathematical morphology operator, which is top-hat transform and morphology opening: (**a**) green channel of the image, (**b**) histogram of the grayscale distribution of the green image, (**c**) result of the direct WTH, (**d**) histogram of grayscale distribution after WTH.

**Figure 7 diagnostics-13-00342-f007:**
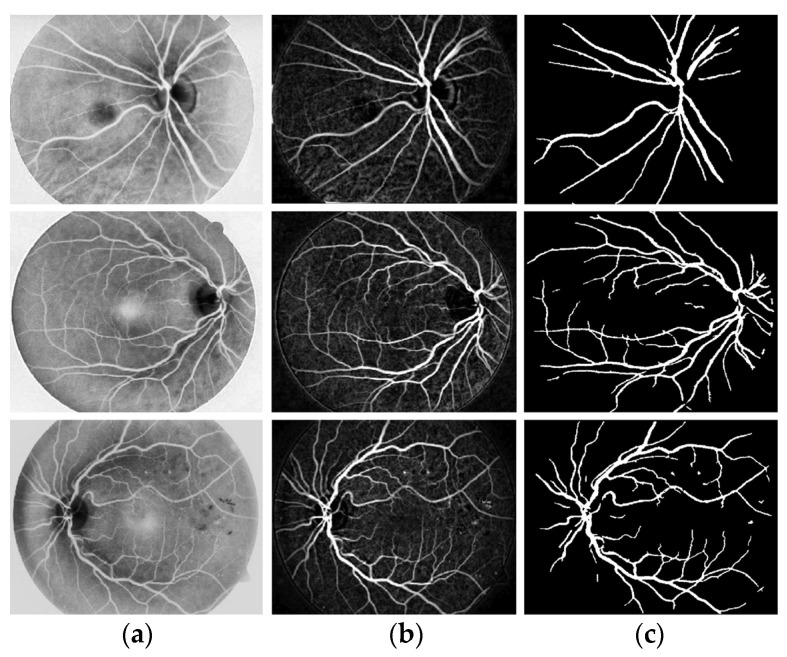
Examples of retinal blood-vessel extraction: (**a**) the result of the direct WTH, (**b**) retinal blood vessels extracted by the proposed WKFM, (**c**) the image after intensity thresholding. From the **top** (DRIVE), **middle** (STARE), and **bottom** (DiaretDB0).

**Figure 8 diagnostics-13-00342-f008:**
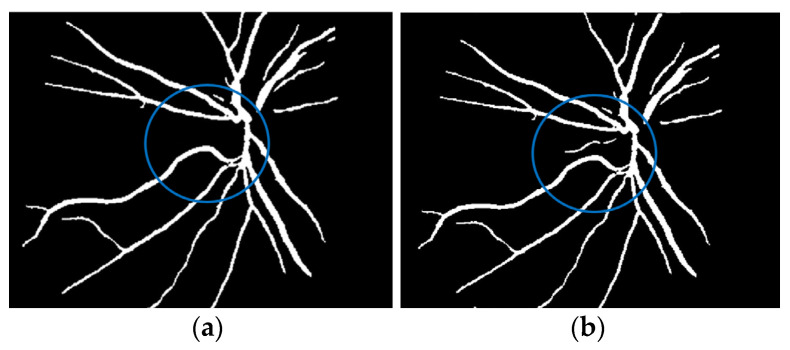
Comparison of the extraction results with WKFCM false extraction and adding morphological dilation operator: (**a**) intensity thresholding with a loose extraction using WKFCM, (**b**) intensity thresholding using the proposed Dilatation-Based Function technique (demonstrated within the blue line).

**Figure 9 diagnostics-13-00342-f009:**
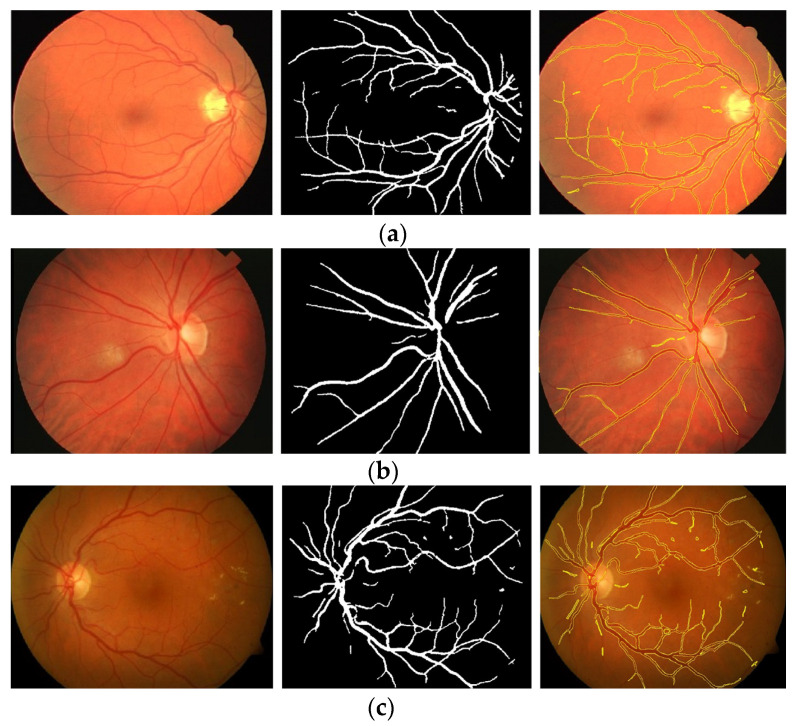
Cross-dataset extraction result comparison of the retinal images from three datasets: (**a**) DRIVE dataset (**first row**), (**b**) STARE dataset (**second row**), (**c**) DiaretDB0 dataset (**third row**).

**Table 1 diagnostics-13-00342-t001:** The dataset information for the extraction of blood vessels.

Datasets	Dimension in Pixels	Number of Images	Training Set (30%)	Test Set (70%)
DRIVE	768 × 584	40	12	28
STARE	605 × 700	80	6	18
DiaretDB0	1500 × 1152	130	39	121

**Table 2 diagnostics-13-00342-t002:** Performance of WKFACM-DBF method with preprocessing and without preprocessing for blood-vessel extraction based on the DRIVE, STARE, and DiaretDB0 datasets.

**Datasets**	**Non-Preprocessing Method**		
	**SEN (%)**	**SPEC (%)**	**ACC (%)**
DRIVE	88.19	88.89	88.46
STARE	88.10	88.41	88.32
DiaretDB0	87.86	88.10>	88.04
**Datasets**	**Preprocessing Method**		
	**SEN (%)**	**SPEC (%)**	**ACC (%)**
DRIVE	98.12	98.20	98.16
STARE	98.42	98.80	98.51
DiaretDB0	98.89	98.10	98.09

**Table 3 diagnostics-13-00342-t003:** Performance of standard FCM and WKFACM-DBF methods for blood-vessel extraction based on the DRIVE, STARE, and DiaretDB0 databases.

**Datasets**	**Standard FCM**		
	**SEN (%)**	**SPEC (%)**	**ACC (%)**
DRIVE	71.75	70.98	71.82
STARE	71.55	70.64	71.52
DiaretDB0	70.59	70.24	70.36
**Datasets**	**WKFACM-DBF**		
	**SEN (%)**	**SPEC (%)**	**ACC (%)**
DRIVE	98.12	98.20	98.16
STARE	98.42	98.80	98.51
DiaretDB0	98.89	98.10	98.09

**Table 4 diagnostics-13-00342-t004:** Performance of WKFCM-DBF method for retinal blood-vessel extraction on the private dataset.

**Images**	**Non-Preprocessed Image**		
	**SEN (%)**	**SPEC (%)**	**ACC (%)**
0001	71.08	70.98	71.10
0002	70.82	70.12	70.54
0004	69.88	70.10	69.91
0005	70.11	69.98	70.05
0006	71.56	71.62	71.58
…	…	…	…
1259	70.36	70.38	70.35
1260	71.89	71.51	71.52
Average	70.59	70.24	7.036
**Images**	**Preprocessed Image**		
	**SEN (%)**	**SPEC (%)**	**ACC (%)**
0001	97.80	97.10	97.12
0002	96.20	96.56	96.48
0004	97.09	97.89	97.44
0005	97.09	97.80	97.37
0006	96.44	96.98	96.12
…	…	…	…
1259	97.18	97.60	97.41
1260	96.78	96.65	96.74
Average	97.38	97.60	97.42

**Table 5 diagnostics-13-00342-t005:** The performance comparison of the WKFACM-DBF method with the usual state-of-the-art methods on the DRIVE dataset.

Authors	Methods	SEN (%)	SPEC (%)	ACC (%)
K. Rezaee et al. [12]	Adaptive filtering, fuzzy entropy, and skeletonization	71.89	97.93	94.63
X. You et al. [18]	Radial projection and semi-supervised approach	74.10	97.51	94.34
J. Mo and L. Zhang [19]	Multi-level deep supervised networks	77.79	97.80	95.21
Z. Jiang et al. [22]	Fully convolutional network with transfer learning	**98.25**	75.40	96.24
W. Li et al. [23]	Active learning method.	77.01	**98.83**	96.92
İ. Atli and O. S. Gedik [24]	Deep learning (Sine-Net)	82.60	**98.24**	96.85
A. P. Lopes et al. [33]	Dilated convolutions	79.03	98.13	95.67
T. A. Soomro et al. [34]	Deep convolutional neural network	80.20	97.40	95.90
Z. Luo et al. [35]	Attention-Dense-U-Net	80.75	98.14	96.63
K. Li et al. [40]	Fully Attention-based Networks	81.45	**98.83**	97.69
**This study**	**WKFCM-DBF**	**98.12**	**98.20**	**98.16**

**Table 6 diagnostics-13-00342-t006:** Extraction results for three datasets: DRIVE, STARE, and DiaretDB0.

Datasets	Original Images	WTH Images	Extracted Images	Superimposed Images
DRIVE	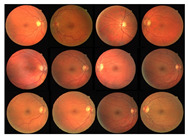	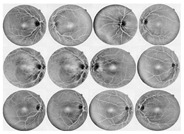	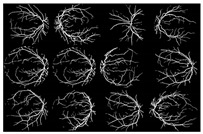	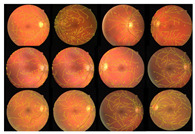
STARE	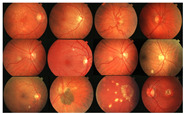		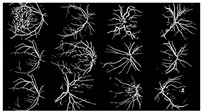	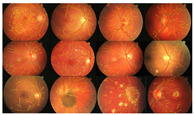
DiaretDB0	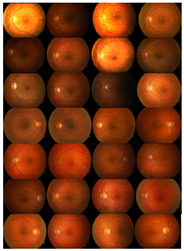		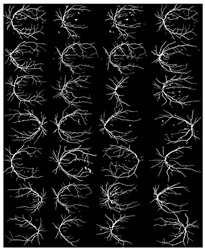	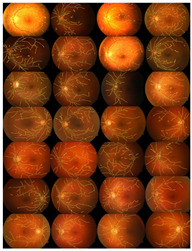

## Data Availability

Not applicable.

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
