# Peer review of "Retinal Blood-Vessel Extraction Using Weighted Kernel Fuzzy C-Means Clustering and Dilation-Based Functions"

_diagnostics, 2023, doi:10.3390/diagnostics13030342_

Round 1
Reviewer 1 Report
The paper presents an interesting idea but should be modified in some of its parts:
- A section on the state of the art is missing even if some references have been inserted in the introduction;
- Figure 1 is very important as it presents the proposed framework. It should be improved in terms of quality and better contextualized and described, step by step, in the text;
- Section 3.2 should be written in algorithmic form and not as a bulleted list. Furthermore, the description in the text should be referenced against the proposed algorithm;
- Missing details about the implementation (software, language, used libraries, etc);
- Performance measures more sensitive to data imbalances such as balanced accuracy or Matthew's correlation coefficient should be used;
- In the specific context the following paper should be cited which uses a segmentation approach to extract structure information from the image:
Manzo, Mario, and Simone Pellino. "FastGCN+ ARSRGemb: a novel framework for object recognition." Journal of Electronic Imaging 30.3 (2021): 033011.
Author Response
Response Letter
|
Comment |
Response |
|
Point 1: - A section on the state of the art is missing even if some references have been inserted in the introduction; |
We have added details of the state of the art, as shown in subsection 1.1 |
|
Point 2: Figure 1 is very important as it presents the proposed framework. It should be improved in terms of quality and better contextualized and described, step by step, in the text; |
Thank you so much Prof. We have added detail of the proposed method step by step in sections 2.1-2.5. |
|
Point 3: Missing details about the implementation (software, language, used libraries, etc); |
We have added detail of software, language, and used data in section 4.5. |
Thank you very much for your kind consideration. We look forward to hearing from you soon.
Warmest regards,
Assoc. Prof. Dr Kittipol Wisaeng, Ph.D.
Electrical and Computer Engineering
Reviewer 2 Report
In this work, the author proposed a three-combination stage method for retinal blood vessel segmentation, including segmentation from background with the Weighted Kernel Fuzzy C-means clustering (WKFCM) method, Dilation-Based Functions (DBF) for enriching details, and threshold method in the conventional segmentation analysis. Overall, the proposed method is similar to the classic segmentation method, and not much new discovery was reported in this manuscript. There are some points need to be further clarified. I recommend a major revision on this manuscript.
1. The image qualities are poor throughout the manuscript. High resolution images are suggested.
2. Many grammar errors need to be corrected.
3. It is not clear why the author used only 30% dataset for training, while 70% dataset for testing, especially considering the small datasets that were used.
4. The standard ‘Accuracy’ is not defined as ‘(sensitivity + specificity)/2’. The author needs to double check the definition.
5. It is more interesting to show connectivity among the segmented images, rather than simply showed ‘accuracy’ in this manuscript.
Author Response
Response Letter
|
Comment |
Response |
|
Point 1: The image qualities are poor throughout the manuscript. High resolution images are suggested. |
Revised |
|
Point 2: Many grammar errors need to be corrected. |
Improved |
|
Point 3: It is not clear why the author used only 30% dataset for training, while 70% dataset for testing, especially considering the small datasets that were used. |
Most research studies are trained with 70% visual data and the remaining 30% for testing. Therefore, the experimental results are promising when using a lot of images in the training phase, and only 30% of the images are left in the experiment. So, the challenge of the proposed method is to test its effectiveness in the proposed way by reducing the number of images to train to 30% and the remaining 70% to use for testing. |
|
Point 4: The standard ‘Accuracy’ is not defined as ‘(sensitivity + specificity)/2’. The author needs to double check the definition. |
Revised in the main text with yellow highlights |
|
Point 5: It is more interesting to show connectivity among the segmented images, rather than simply showed ‘accuracy’ in this manuscript. |
Revised section 4.2 |
Thank you very much for your kind consideration. We look forward to hearing from you soon.
Warmest regards,
Assoc. Prof. Dr Kittipol Wisaeng
Reviewer 3 Report
Manuscript Title “Retinal Blood Vessel Segmentation Using a Weighted Kernel Fuzzy C-Means Clustering and Dilation-Based Functions”
General comment:
FCM is a very useful technique in dealing of the fuzzy imaging, however, the way that in author’s writing makes it very difficult to follow, especially the definition of variables or parameters. The green filter as author denoted can have the optimal image in judging, so green filter is fixed in this study or it still changes according to different raw images as well. The detail comments on some specific points are listed below,
Specific comment:
1. L216, Figure 1 does not show a clear vision of how the technique going to function, why there are two blocks of pre-processing images with same function.
2. The definition of tau in eq. 1 is unclear and the correlation between tau and H (entropy) in eq. 2 is also unclear too.
3. The kernel of fuzzy theory is exactly created on the definition of eq. 1, thus, it needs to be further elaborated
4. The definition of eq. 4 is unclear to the reader, especially the gamma value
5. L268, the radius of 7?? The unit is pixel or mm or what??
6. Eq. 7, uik = [the denominator term, xk- vj or vi ??] should be i to consistent with the summation_i=1; and vi is actually the weighted average
7. L319, the common , should be replaced by solid dot ·
8. Eq. 13. 14 similar to eq. 7 and proved that expression in eq. 7 is wrong
9. L455, the definition of TP, FN, TN, and FP as well as the sensitivity, specificity and accuracy is a well-known common sense or defined just by the authors themselves??
10. L471, table 2, what is the difference between the upper and lower parts in each table? In contrast, table 3 has clear definition between upper and lower part (standard FCM and WKFACM-DBF), plus lower part of either table 2 or 3 is the same, it is redundant at all, so confusion indeed.
11. The table 5 is very persuadable, try to elaborate it more in the discussion to enrich the contribution of the FCM feature in reality.
Author Response
Response Letter
|
Comment |
Response |
|
Point 1: Figure 1 does not show a clear vision of how the technique going to function, why there are two blocks of pre-processing images with same function |
Revised in Fig. 1 |
|
Point 2: The definition of tau in eq. 1 is unclear and the correlation between tau and H (entropy) in eq. 2 is also unclear too. |
Revised in section 2.1. |
|
Point 3: The kernel of fuzzy theory is exactly created on the definition of eq. 1, thus, it needs to be further elaborated. |
|
|
Point 4: The definition of eq. 4 is unclear to the reader, especially the gamma value. |
|
|
Point 5: the radius of 7?? The unit is pixel or mm or what?? |
Revised |
|
Point 6: Eq. 7, uik = [the denominator term, xk- vj or vi ??] should be i to consistent with the summation_i=1; and vi is actually the weighted average |
|
|
Point 7: L319, the common , should be replaced by solid dot · |
Revised |
|
Point 8: Eq. 13. 14 similar to eq. 7 and proved that expression in eq. 7 is wrong |
Removed Eq. 7 |
|
Point 9: L455, the definition of TP, FN, TN, and FP as well as the sensitivity, specificity and accuracy is a well-known common sense or defined just by the authors themselves?? |
We added detail of sensitivity, specificity and accuracy in section 4.2. |
|
Point 10: L471, table 2, what is the difference between the upper and lower parts in each table? In contrast, table 3 has clear definition between upper and lower part (standard FCM and WKFACM-DBF), plus lower part of either table 2 or 3 is the same, it is redundant at all, so confusion indeed. |
Revised in Table 2. |
|
Point 10: The table 5 is very persuadable, try to elaborate it more in the discussion to enrich the contribution of the FCM feature in reality. |
Revised in section 4.5. |
Thank you very much for your kind consideration. We look forward to hearing from you soon.
Warmest regards,
Assoc. Prof. Dr Kittipol Wisaeng
Round 2
Reviewer 1 Report
The suggested citation is missing:
Manzo, Mario, and Simone Pellino. "FastGCN+ ARSRGemb: a novel framework for object recognition." Journal of Electronic Imaging 30.3 (2021): 033011.
Author Response
Response Letter
|
Comment |
Response |
|
Point 1: Manzo, Mario, and Simone Pellino. "FastGCN+ ARSRGemb: a novel framework for object recognition." Journal of Electronic Imaging 30.3 (2021): 033011. |
In this article, there is not much material related to the use of CNNs, however, we will definitely apply and refer to them in the future articles. |
Thank you very much for your kind consideration. We look forward to hearing from you soon.
Warmest regards,
Assoc. Prof. Dr Kittipol Wisaeng, Ph.D.
Electrical and Computer Engineering
Reviewer 2 Report
The revision looks good to me. I recommend to accept this manuscript.
Author Response
Response Letter
|
Comment |
Response |
|
Point 1: The revision looks good to me. I recommend to accept this manuscript.
|
Thank you so much Prof. |
Thank you very much for your kind consideration. We look forward to hearing from you soon.
Warmest regards,
Assoc. Prof. Dr Kittipol Wisaeng, Ph.D.
Electrical and Computer Engineering